# A Nomogram Predicting Progression Free Survival in Patients with Gastrointestinal Stromal Tumor Receiving Sunitinib: Incorporating Pre-Treatment and Post-Treatment Parameters

**DOI:** 10.3390/cancers13112587

**Published:** 2021-05-25

**Authors:** Yau-Ren Chang, Wen-Kuan Huang, Shang-Yu Wang, Chiao-En Wu, Jen-Shi Chen, Chun-Nan Yeh

**Affiliations:** 1Department of Surgery and GIST Team, Chang Gung Memorial Hospital at Linkou, ChangGung University College of Medicine, Taoyuan 33305, Taiwan; eq99887766@hotmail.com (Y.-R.C.); shangyuwang@gmail.com (S.-Y.W.); 2Division of Hematology-Oncology, Department of Internal Medicine and GIST Team, Chang Gung Memorial Hospital at Linkou, Chang Gung University College of Medicine, Taoyuan 33305, Taiwan; medfoxtaiwan@gmail.com (W.-K.H.); jiaoen@gmail.com (C.-E.W.); js1101@cgmh.org.tw (J.-S.C.)

**Keywords:** sunitinib, gastrointestinal stromal tumor, prognostic nomogram model, KIT genotype, hypertension, hand–foot skin reaction, survival

## Abstract

**Simple Summary:**

Sunitinib has been approved as the second-line targeted treatment for gastrointestinal stromal tumor (GIST) after imatinib failure. It is thus necessary to effectively assess prognosis after sunitinib use. However, the current assessment remains insufficient for the contemporary period. We examined prognostic factors influencing progression-free survival. Furthermore, we constructed a prognostic nomogram model using these significant pre-treatment and post-treatment variables.

**Abstract:**

The present study aimed to construct a prognostic nomogram incorporating pre-treatment and post-treatment factors to predict progression-free survival (PFS) after use of sunitinib in patients with metastatic gastrointestinal stromal tumors (GISTs) following imatinib intolerance or failure. From 2007 to 2018, 109 metastatic GIST patients receiving sunitinib at Chang Gung Memorial Hospital, Taiwan, were enrolled. A prognostic nomogram to predict PFS was developed. Sixty-three male and forty-six female metastatic GIST patients, with a median age of 61 years (range: 15–91 years), received sunitinib. The median PFS for 109 patients is 9.93 months. For pre-treatment factors, male gender, body mass index more than 18.5 kg/m^2^, no sarcopenia status, higher lymphocyte count, lower platelet/lymphocyte ratio, good performance status, higher sunitinib dose, and non-liver metastasis were significantly associated with favorable PFS. For post-treatment factors, adverse events with hypertension, hand–foot skin reaction, and diarrhea were significantly associated with favorable PFS. However, only eight clinicopathological independent factors for PFS prediction were selected for prognostic nomogram establishment. The calibration curve for probability of PFS revealed good agreement between the nomogram prediction and actual observation. High risk patients will experience the lowest PFS. A prognostic nomogram integrating eight clinicopathological factors was constructed to assist prognostic prediction for individual patients with advanced GIST after sunitinib use.

## 1. Introduction

Gastrointestinal stromal tumors (GISTs) arise from mesenchymal tissue in the gastrointestinal (GI) tract and peritoneum, accounting for the most common mesenchymal malignancy of the GI tract [1]. Curative surgical resection provides chance of cure and remains the standard of treatment for GISTs. Nonetheless, postoperative recurrence is not uncommon [2].

GISTs have been reported to originate from the interstitial cells of Cajal, expressing transmembranous KIT receptor with tyrosine kinase activity [3]. Gain-of-function mutations of KIT in GISTs lead to constitutive and persistent activation of KIT signaling, leading to aberrant cell proliferation and resistance to apoptosis [4].

Imatinib mesylate (IM) is a selective tyrosine kinases inhibitor, including KIT and platelet-derived growth factors (PDGFRs), showing a promising clinical outcome for a patient with an advanced GIST [5] and has been established as the standard first-line therapy [5,6,7,8,9]. However, progression of GIST upon IM treatment inevitably develops within two to three years [8,9].

Sunitinib is an oral multi-targeted tyrosine kinase inhibitor against KIT, and PDGFRs), glial cell line-derived neurotrophic factor receptor (rearranged during transfection; RET), vascular endothelial growth factor receptors (VEGFRs), colony-stimulating factor 1 receptor (CSF-1R), and FMS-like tyrosine kinase-3 receptor (FLT3) [10,11,12,13,14,15]. Sunitinib has been approved for the standard second-line treatment of GIST after failure of IM for a decade [16].

Prognostic nomograms have been developed for several types of malignancy [17,18,19]. The nomogram models have been proposed as alternatives or even as new standards due to their comparable ability with traditional staging systems [20,21,22]. The present study aimed to establish a prognostic nomogram incorporating pre-treatment post-treatment factors and to predict PFS for advanced GIST patients receiving sunitinib after IM intolerance or failure.

## 2. Materials and Methods

### 2.1. Patients

We retrospectively reviewed 299 patients with recurrent, unresectable, or metastatic GISTs, which were histologically confirmed by expression of CD117 or DOG1. They were treated at the Department of Medical Oncology and Surgery, Chang Gung Memorial Hospital, Linkou, between 2007 and May 2018. We included patients who demonstrated IM failure by disease progression (based on Response Evaluation Criteria in Solid Tumors (RECIST)) [23] or discontinuation of IM due to toxicity. Patients with Eastern Cooperative Oncology Group (ECOG) performance statuses of 0 to 3 and adequate cardiac, hepatic, renal, coagulation, and hematologic function were required (Appendix A). We excluded patients with lack of recovery from the acute toxic effects of previous anticancer therapy or IM treatment, discontinuation of IM within 2 week or of any other approved or investigational drug for GIST within 4 week before starting sunitinib treatment, clinically significant cardiovascular events or disease in the previous 12 months, diabetes mellitus with clinical evidence of peripheral vascular disease or diabetic ulcers, or a diagnosis of any second malignancy within the previous 5 years. Patients were allowed to have had previously chemotherapeutic treatment (the last chemotherapy treatment must have been at least 4 week before study entry) and undergone radiotherapy or surgery, or both. The study was approved by the institutional review board of Chang Gung Memorial Hospital. The written informed consent for drug administration and the analysis of tumor-associated genetic alteration was obtained independently from each patient.

### 2.2. Study Design and Follow-Up Study

We conducted a retrospective analysis to evaluate the effectiveness on prognosis and safety of sunitinib in Taiwanese GIST patients. Patients were administered daily 50 mg (4 week on, 2 week off) or 37.5 mg continuously of sunitinib using 12.5 mg capsules orally with food. The two treatment schedules showed similar survival efficacies in previous trials (Appendix A). Patients had regular physical examinations and performance status, body weight, differential blood count, and serum biochemistry were evaluated. The administration of each dose and any adverse events were recorded for each patient. Computed tomography (CT) was performed every 3 months for the first 3 years and every 6 months for the following 2 years to assess patient response. We measured objective tumor assessments using Response Evaluation Criteria in Solid Tumors (RECIST) with a minor modification to allow use of standard radiographic protocols for spiral CT [23]. We defined progression-free survival (PFS) as no progression after sunitinib use. Overall survival (OS) was defined as survival after sunitinib use until death. Safety and tolerability were assessed by analysis of adverse events, physical examinations, vital signs, ECOG performance status, and laboratory abnormality assessments, including complete blood count with differential count, serum electrolyte measurements, and electrocardiogram.

Toxic effects were categorized into hematological and non-hematological adverse effects and recorded in accordance with the National Cancer Institute Common Toxicity Criteria [24]. The clinical response to TKI was assessed by CT with the criteria of the RECIST 1.1 [24].

### 2.3. Analysis of KIT and PDGFRA Mutations

The specimen from biopsy or surgical resection of GIST with progression after imatinib use was formalin-fixed and paraffin embedded. Sections from formalin-fixed, paraffin embedded pretreatment specimens were trimmed to enrich tumor cells. Then, we performed polymerase chain reaction amplification of genomic DNA for KIT and PDGFRA to analyze the mutations, as in previously published studies [8,9].

### 2.4. Definition of Sarcopenia

A diagnosis sarcopenia was made by muscle mass assessment with CT scan [25]. The psoas muscle index at level of the third lumbar vertebra (L3-PMI) has been considered representative for skeletal muscle mass generality [26]. A cross-section area of L3-PM was obtained from picture archiving and communication system (PACS) and quantified based on Hounsfield unit thresholds (−29 to +150), and subsequently measured by IMAGEJ processing system [27] (Appendix A) and normalized by patient height (L3-PMI = total psoas area (TPA)/height2, mm^2^/m^2^). A PMI cut-off value for sarcopenia definition was according to a previous study regarding Asian adults (636 mm^2^/m^2^ for males and 392 mm^2^/m^2^ for females) [26]. To avoid estimated errors, an exclusive survey was used and all images were viewed twice and resultant data were from averaged scales.

### 2.5. Nomogram Creation

A nomogram was analyzed by R software (version 2.14.1) with the rms package and other dependent packages (http://www.r-project.org/, accessed on 10 September 2020). We used a categorical variable based on the result of receiver operating characteristic (ROC) curve. We used the concordance index (C-index) to measure the performance of the nomogram. Calibration curve was plotted by comparing nomogram predicted versus observed probability of survival. For internal validation, bootstrapping with 1000 resamples was used.

### 2.6. Statistical Analysis

Categorical variables were compared using the Chi-square test or Fisher’s exact test accordingly (based on expected values individually). Continuous variables were compared using the t-test or Mann–Whitney U-test according to distribution of data. Survival curves were depicted by the Kaplan–Meier curve and the log-rank test was applied for statistical comparison. Cox regression analysis was used for multivariate analyses and to formulate a nomogram. To analyze the nomogram points as the prognostic factors for recurrence, we used recursive partitioning analysis (PRA), a statistical methodology that creates a survival analysis tree, to establish an optimal cut-off point that better predicts the recurrence. This statistical approach for survival tree development was used previously [28].

## 3. Results

### 3.1. Clinicopathological Characteristics of Patients

The demographic features of 109 GIST patients receiving sunitinib are shown in Table 1. There were 63 male and 46 female metastatic GIST patients with a median age of 61 years (range: 15–91 years). The median follow-up time in this cohort was 22.4 months (range = 1.2–142.5) with the median PFS being 9.9 months and OS being 31.8 months, respectively (Figure 1A,B).

Eighty (73.4%) of the 109 GIST patients with IM failure or intolerance had tumor samples suitable for genetic analysis. Fifty (63.8%) of the 80 GISTs had activated mutations of KIT exons 9 and 11. Twenty-four patients (24/80; 30%) developed secondary mutations when they used sunitinib. Fourteen patients had a missense mutation in exon 13 and the other ten patients had one in exon 17. Patients harbored exon 11 mutations with secondary mutations being more frequent than exon 9 mutations (22/24 (91.6%) versus 2/24 (8.4%) (Table 1). Anemia was the most common grade III adverse effect (19.2%), followed by hand foot skin reaction (14.7%) (Appendix A). The median PFS for the 14 patients with tumors harboring secondary mutations in exon 13 was 23.3 months. While the median PFS for the 10 patients with tumors harboring secondary mutations in exon 17 was 8.3 months (Table 2).

### 3.2. Independent Prognostic Factors in the Training Cohort

Table 2 summarizes the univariate analysis regarding pre- and post-treatment parameters. The univariate analysis revealed 11 significant prognostic factors, including gender, body mass index, sarcopenia, lymphocyte count, platelet/lymphocyte ratio, ECOG, sunitinib dose, metastatic site, hypertension, hand–foot skin reaction, and diarrhea.

We further created the nomogram using Cox proportional model analysis with eight significant parameters, including gender, ECOG, platelet/lymphocyte ratio, sarcopenia, metastatic site, sunitinib dose, hypertension, and hand–foot skin reaction. (Table 3, Appendix A and Figure 2A).

### 3.3. Prognostic Nomogram for PFS and Risk Strastification

A nomogram was constructed based on the results of the final multivariable model (Figure 2A). The concordance index for the model was. 0.77 (95% CI: 0.73–0.81) between nomogram prediction and actual observation. The calibration curve for the probability of survival at 1,2 and 3 years after sunitinib use revealed a good agreement between the nomogram prediction and actual observation (Figure 2B).The formula (Table 4) included gender (male: 0 points, female: 68 points), ECOG (0:0 points, 1/2: 14 points, 3: 98 points), platelet/lymphocyte ratio (≤270:0 points, above 270:48 points), sarcopenia (presence: 77 points, absence: 0 point), metastatic site (non-liver metastasis: 0 points, liver metastasis: 74 points, IV: 100 points), sunitinib dose (25 mg daily: 72 points, 37.5 mg daily: 81 points, 50 mg daily: 0 points), hypertension (absence: 78 points, presence: 0 points), and hand–foot skin reaction (absence: 100 points, presence: 0 points). 22 patients in the lower quartile (total points 0–227) had a significantly better prognosis with a median survival of 61.0 months compared to any other quartile (95% CI 0.1–146.2, *p* < 0.001); on the opposite, 36 patents with a higher score >385 had a median survival of 3.3 months (95% CI 2.6–4.0) (Figure 3A,B).

## 4. Discussion

Our study demonstrated a comparable efficacy for survival and safety of sunitinib to previous studies after imatinib failure [29,30,31,32]. Moreover, we constructed a clinical risk model and nomogram for prediction of PFS. We identified unfavorable factors such as being female, ECOG score of 3, liver metastasis, the presence of sarcopenia, and platelet/lymphocyte ratio >270. Treatment-related hypertension and hand-foot skin reaction were associated with favorable prognosis. Of note, we constructed this prognostic model based on the whole cohort. We did not split the cohort into the training set and test set due to limited sample size. Therefore, further external validation is needed to confirm our results.

In our study, the survival outcome of second-line sunitinib use was better than that reported by the prior clinical trial. The phase III trial demonstrated median PFS and OS of approximately 6–7 months and 18 months [16,33]. The early real-world studies also showed similar survival efficacy. A worldwide treatment-use study reported median PFS and OS of 8.3 and 16.6 months, respectively [29]. A Korean study also found a consistent efficacy (median PFS and OS of 7.1 and 17.6 months, respectively) [30]. In contrast, our study’s results were approximately 2–3 months and 13 months longer than those reported in these early studies (median PFS and OS of 9.9 and 31.8 months, respectively). These early studies were conducted earlier than 2010, while our study collected patients between 2007 and 2018. A recent study enrolled 91 patients between 2005 and 2015 demonstrated median PFS and OS of 8.8 and 27.5 months, respectively), which was in line with our study [32]. The reason for the survival difference remains to be determined, while available regorafenib after sunitinib failure, flexible sunitinib dosing, and less discontinuation due to adverse events may lead to prolonged survival [34].

Sarcopenia defined as loss of muscle mass combined with a decrease in muscle strength and physical performance is an important prognostic factor in gastrointestinal cancers [35]. Pre-imatinib sarcopenia has been found to predict imatinib-related toxicities in patients with advanced GIST [36]. Six-month imatinib treatment reversed sarcopenia in seven patients (63.6%). In our study, we found that pre-sunitinib sarcopenia is an independent prognostic factor for PFS, while sarcopenic reversal was not significantly associated with a better prognosis. Our results, together with a previous study, suggested that the presence of sarcopenia before TKI use is important for prediction of prognosis and treatment-related adverse events. Moreover, sarcopenia reversal by TKI may benefit patients with GIST, while further prospective studies are needed to confirm these findings.

We found that the frequencies of non-hematologic toxicities were similar to those reported in previous studies [29]. Of note, the most common grade 3–4 adverse event is hand–foot skin reaction, which was consistent with the worldwide study and an Asian study [29,30]. Several Asian studies found higher hematologic toxicities (69–90%) than those (18–57%) in global trials [30,37]. However, our Taiwanese population reported the frequencies of anemia (64.2%), thrombocytopenia (24.8%), and leukopenia (26.6%), which were not different from those from global population [29]. Whether there is ethnic difference in terms of sunitinib-related adverse events needs further pharmacogenetic investigation.

Hypertension was a significantly prognostic factors for PFS in our study. A retrospective analysis of advanced GIST studies also demonstrated that sunitinib-associated hypertension correlated with response rate, PFS, and OS [38]. Inhibition of VEGFR-2 by sunitinib increases peripheral vascular resistance, which can lead to the development of hypertension. Therefore, hypertension is thought an on-target effect of sunitinib, predicting the treatment efficacy [38,39].

The presence of hand–foot skin reaction upon sunitinib use associated with longer PFS has been demonstrated in patients with metastatic renal cell carcinoma [40,41], which was in line with our results in advanced GIST. Although the mechanism of sunitinib leading to hand–foot skin reaction remains unknown, inhibition of VEGFR has been recognized as an important factor of pathogenesis of hand–foot skin reaction from multi-targeted TKIs [42]. A recent study mechanistically demonstrated the activation of EGFR on keratinocytes by soluble heparin-binding epidermal growth factor released from vascular endothelial cells, promoting the development of sorafenib-associated keratinization [43]. The aforementioned results suggest that loss of vascular competence is important in the pathogenesis of TKI-related hand–foot skin reaction.

The most common resistant mechanism to imatinib is secondary mutations at KIT exon 17 and exon 13, but not downstream signaling or other signaling, which is stunning as it underscores the unique role of KIT in oncogene addiction in GIST [44]. The frequency of secondary mutations is associated with the location of the primary KIT mutations. GISTs harboring primary KIT exon 11 mutations more commonly developed secondary KIT mutations (46–61%) as compared with primary exon 9 (0–15%) [45,46]. Consistent with these results, our study found that secondary mutations in 91.6 (22/24) and 8.4% (2/24) of primary KIT exon 11 and exon 9 mutations, respectively. The mechanisms for the different frequencies between exon 11 and exon 9 remain unclear. Patients with GISTs harboring an exon 11 mutation have longer duration of imatinib treatment than those harboring an exon 9 mutation, suggesting the development of a secondary mutation is associated with duration of imatinib treatment. Therefore, selective pressure resulting in resistant clones upon imatinib treatment might, in part, explain these observations.

Several studies consistently found that patients with KIT exon 9 mutations present with longer survival of sunitinib treatment than those with KIT mutations in exon 11 [29,30,47]. Secondary mutations also influence the efficacy of sunitinib [46,48]. Patients with secondary KIT exon 13 and exon 14 mutations had longer PFS and OS than those with KIT 17 or 18 mutations, which confer resistance to sunitinib [46]. While there was no statistical significance identified, our results also found that patients with secondary mutation of exon 13 had higher rates of 1-year and 3-year PFS than those with secondary mutations of non-exon 13. Although stratification by mutational status is promising to guide sunitinib treatment, complicated inter- and intra-lesion genetic heterogeneity of resistant tumors is present [49]. As a result, one single biopsy may not be representative. Future studies are worthwhile to characterize the impact between predominant and minor mutations on sunitinib-associated prognosis.

## 5. Conclusions

In conclusion, our results demonstrated the efficacy and safety profile of sunitinb in a contemporary period (2007–2018). With comprehensive analysis of demographics, tumors, and biochemical characteristics, we recognized several predictive factors for PFS. We specifically confirmed that sarcopenia before sunitinib use is an independent prognostic factor. Moreover, we constructed a prognostic model, a nomogram, for GIST patients receiving sunitinib, while further studies or external validation is needed to verify these results.

## Figures and Tables

**Figure 1 cancers-13-02587-f001:**
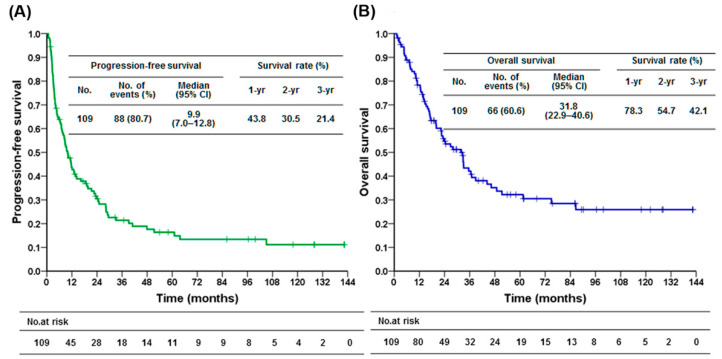
Survival after sunitinb use in patients with GIST. (**A**) Kaplan–Meier plot of the progression-free survival (PFS) for 109 patients with advanced gastrointestinal stromal tumor (GIST) treated with sunitinib. (**B**) Kaplan-Meier plot of the overall survival (OS) for 109 patients with advanced gastrointestinal stromal tumor (GIST) treated with sunitinib.

**Figure 2 cancers-13-02587-f002:**
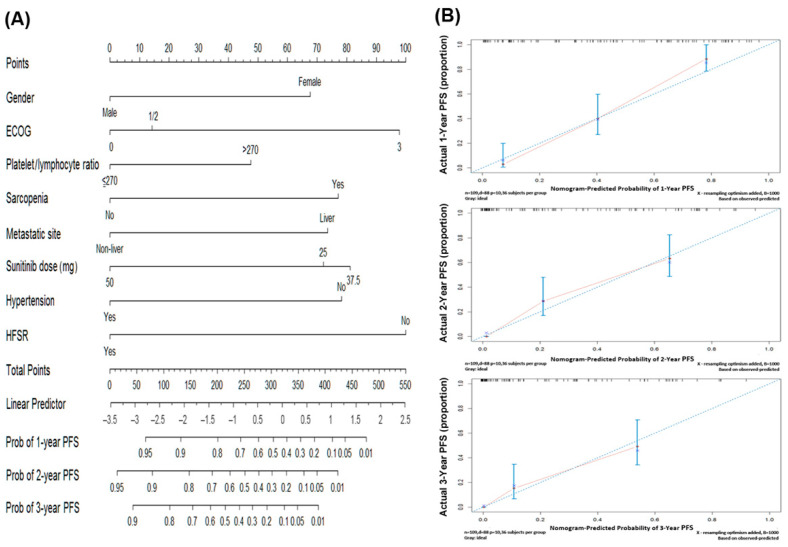
Nomogram and calibration plots: (**A**) Nomogram based on 109 patients with advanced GISTs treated with sunitinib. (**B**) Calibration curves with nomogram predicted 1-year, 2-year, and 3-year PFS and actually observed survival.

**Figure 3 cancers-13-02587-f003:**
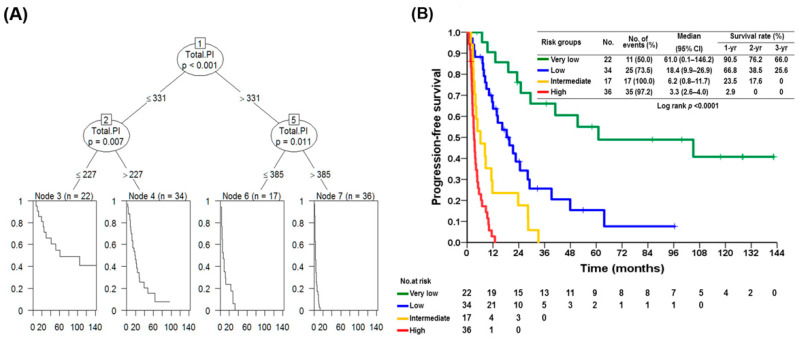
Survival analysis tree and Kaplan–Meier plot by nomogram points. (**A**) A survival analysis tree was used and to establish an optimal cut-off point to better predict the recurrence. (**B**) Kaplan–Meier plot of the PFS of 109 patients with advanced GISTs treated with sunitinib in terms of risk with different scores.

**Table 1 cancers-13-02587-t001:** The demographic characteristics of 109 advanced GIST patients under Sunitinib therapy.

Variables	No.	Percentage (%) or Mean ± SD
**Basic data**		
Gender		
Male/Female	63/46	57.8/42.2
Age (years)		60.9 ± 13.6
ECOG when start		
0/1/2	24/47/22	22.0/43.1/20.2
3	16	14.7
**Body composition**		
Weight (kg)		61.0 ± 12.6
Height (cm)		160.0 ± 8.5
BMI (kg/m^2^)		23.7 ± 4.0
BMI grading (kg/m^2)^		
<18.5	8	7.3
18.5–27	83	76.2
>27	18	16.5
Albumin > 3.5 (g/dL)		
Yes	65	40.4
No	44	59.6
**Tumor characteristics**		
Tumor size (cm)		10.4 ± 6.0
Location		
Stomach	39	35.8
Small bowel	53	48.7
Colorectal	8	7.3
Peritoneum	1	0.9
Other	8	7.3
Liver Metastasis		
Yes/No	70/39	64.2/35.8
**Genetic mutation when using sunitinib (N = 109)**		
Exon 9	15	13.8
Exon 9 and 17	2	1.8
Exon11	36	33.0
Exon 11 and 13	14	12.8
Exon 11 and 17	8	7.3
Wild type	5	4.6
Unknown	29	26.6
**Sunitinib dosage**		
Dosage divided ^b^		
Yes/No	70/39	64.2/35.8
Initial dosage (mg)		
25	17	15.6
37.5	79	72.5
50	13	11.9
Directly shift to sunitinib ^c^		
Yes/No	49/60	45.0/55.0

Note: Abbreviation: GIST: Gastrointestinal Stromal Tumor; SD: Standard Deviation; ECOG, Eastern Clinical Oncology Group performance status; BMI, body mass index. ^b^ Standard dosage of sunitinib (50 mg QD) was given for 4 weeks, followed by a two-week drug-free period. In the study, we divided the dosage into 12.5 mg QID/25 mg BID to reduce the toxicity. ^c^. When the GIST patients experienced disease progression during imatinib use, the patients would receive imatinib escalation or a direct shift to sunitinib.

**Table 2 cancers-13-02587-t002:** Progression-free survival analysis of each predictor variable (Univariate).

Predictor Variables	Median Survival (m) (95% CI)	1-Year PFS (%)	3-Year PFS (%)	*p* Value
Age (years)				0.692
≤61 (*n* = 56)	9.5 (5.5–13.5)	44.1	25.3	
>61 (*n* = 53)	10.2 (6.4–14.0)	43.5	16.0	
Gender				0.024
Male (*n* = 63)	12.9 (3.3–22.5)	52.4	29.6	
Female (*n* = 46)	8.3 (5.8–10.8)	32.3	11.5	
Body mass index				0.001
<18.5 (*n* = 8)	3.2 (2.5–4.0)	0	0	
18.5–27 (*n* = 83)	11.4 (6.5–16.3)	46.7	22.3	
>27 (*n* = 18)	11.3 (7.5–15.1)	50.0	26.7	
Sarcopenia				0.005
Yes (*n* = 25)	4.6 (3.3–6.0)	16.8	4.6	
No (*n* = 84)	13.1 (6.3–19.9)	51.6	13.1	
Lymphocyte count				0.008
≤858 (*n* = 29)	6.2 (3.3–9.2)	20.7	10.3	
>858 (*n* = 80)	12.9 (4.5–21.3)	52.5	25.6	
Neutrophil/lymphocyte ratio				0.122
≤2.14 (*n* = 32)	14.3 (6.4–22.2)	58.6	27.4	
>2.14 (*n* = 61)	7.2 (3.8–10.5)	34.4	16.8	
Monocyte/lymphocyte ratio				0.106
≤0.19 (*n* = 33)	12.0 (3.2–20.9)	53.1	34.0	
>0.19 (*n* = 60)	7.4 (5.1–9.7)	35.6	13.7	
Platelet/lymphocyte ratio				0.009
≤270 (*n* = 75)	13.8 (7.1–20.7)	53.4	25.9	
>270 (*n* = 34)	6.2 (2.5–9.9)	23.5	11.8	
ECOG				0.017
0 (*n* = 24)	24.6 (5.2–44.0)	62.5	30.8	
1/2 (*n* = 69)	9.0 (57.3–12.6)	42.8	20.9	
3 (*n* = 16)	3.0 (51.8–4.2)	16.7	8.4	
Albumin (g/dL)				0.343
≤3.5 (*n* = 44)	5.6 (52.2–8.9)	36.7	19.9	
>3.5 (*n* = 65)	11.7 (58.2–15.3)	48.7	22.8	
Direct to Sunitinib				0.223
No (*n* = 49)	8.3 (56.1–10.5)	33.4	13.4	
Yes (*n* = 60)	12.9 (55.1–20.7)	51.6	27.4	
Sunitinib dose (mg)				0.029
25 (*n* = 17)	6.5 (1.3–11.7)	35.3	N/A	
37.5 (*n* = 79)	10.2 (6.3–14.0)	43.5	20.3	
50 (*n* = 13)	19.1 (-)	56.4	47.0	
Primary site				0.658
Stomach (*n* = 39)	11.3 (8.3–14.3)	50.2	23.2	
Small bowel (*n* = 53)	10.2 (4.7–15.6)	44.7	21.6	
Colorectum (*n* = 8)	14.3 (0.1–28.7)	62.5	23.4	
Others (*n* = 9)	8.3 (6.9–9.6)	33.3	11.1	
Metastatic site				0.005
Non-liver (*n* = 39)	22.6 (14.4–30.8)	70.6	34.1	
Liver (*n* = 70)	7.0 (3.4–10.5)	29.1	14.5	
Genetic status (Secondary mutation)				0.232
Exon 13 (*n* = 14)	23.3 (9.0–37.5)	69.6	31.3	
Exon 17 (*n* = 10)	8.3 (0.4–16.2)	50.0	0.0	
Hypertension				0.001
No (*n* = 81)	8.3 (5.3–11.2)	35.7	12.4	
Yes (*n* = 28)	28.3 (17.9–38.7)	67.1	44.7	
Hand–foot syndrome				<0.0001
No (*n* = 66)	5.6 (2.4–8.7)	29.5	13.1	
Yes (*n* = 43)	19.7 (12.6–26.7)	65.0	33.5	
Diarrhea				0.021
No (*n* = 67)	7.4 (3.5–11.4)	33.4	15.1	
Yes (*n* = 42)	19.7 (7.3–32.0)	59.5	30.6	
Fatigue				0.141
No (*n* = 72)	11.4 (7.0–15.9)	47.4	25.5	
Yes (*n* = 37)	9.4 (2.5–16.3)	36.9	14.2	
Anemia				0.548
No (*n* = 39)	11.4 (8.3–14.5)	46.2	17.9	
Yes (*n* = 70)	9.0 (5.1–12.9)	42.7	23.8	
Thrombocytopenia				0.268
No (*n* = 82)	9.4 (7.1–11.7)	41.0	19.2	
Yes (*n* = 27)	13.8 (6.3–21.4)	52.8	28.8	
Leukopenia				0.726
No (*n* = 80)	9.0 (5.7–12.3)	42.7	22.1	
Yes (*n* = 29)	11.6 (7.2–16.0)	46.7	19.4	
Anorexia				0.162
No (*n* = 83)	11.3 (7.0–15.6)	47.1	24.4	
Yes (*n* = 26)	9.4 (3.7–15.1)	32.8	12.3	
Edema				0.053
No (*n* = 90)	10.2 (6.2–14.1)	47.6	24.5	
Yes (*n* = 19)	4.6 (3.2–6.1)	24.6	6.1	
Hepatic toxicity				0.559
No (*n* = 94)	9.5 (6.2–12.8)	42.3	21.3	
Yes (*n* = 18)	16.3 (0.1–35.1)	53.3	22.9	
Hypothyroid				0.152
No (*n* = 104)	9.5 (6.7–12.3)	42.0	19.3	
Yes (*n* = 5)	39.2 (17.3–61.1)	80.0	60.0	

**Table 3 cancers-13-02587-t003:** Multivariate Cox regression model of factors predicting the progression-free survival.

Predictor Variables	Hazard Ratio (HR)	95% CI of HR		*p* Value
		Lower	Upper	
Gender				
Male	1			
Female	2.100	1.270	3.470	0.004
ECOG				
0	1			
1/2	1.168	0.649	2.101	0.604
3	2.922	1.363	6.265	0.006
Platelet/lymphocyte ratio				
≤270	1			
>270	1.687	1.051	2.710	0.030
Sarcopenia				
Yes	2.333	1.251	4.349	0.008
No	1			
Metastatic site				
Non-liver	1			
Liver	2.241	1.369	3.671	0.001
Sunitinib dose (mg)				
25	2.205	0.809	6.008	0.122
37.5	2.437	1.011	5.872	0.047
50	1			
Hypertension				
No	2.361	1.331	4.186	0.003
Yes	1			
Hand–foot syndrome				
No	2.995	1.835	4.888	<0.0001
Yes	1			

**Table 4 cancers-13-02587-t004:** Point assignment for each variable and prognostic score for progression-free survival.

**Predictor Variables**	**Points Assigned**	**Total Point Score**	**Probability of 1-Year PFS**
		476	0.01
Gender		437	0.05
		413	0.10
Male	0	381	0.20
Female	68	354	0.30
ECOG		329	0.40
		304	0.50
0	0	276	0.60
1/2	14	243	0.70
3	98	200	0.80
Platelet/lymphocyte ratio		132	0.90
		66	0.95
≤270	0	Total point score	Probability of 2-year PFS
>270	48	424	0.01
Sarcopenia		385	0.05
		361	0.10
Yes	77	328	0.20
No	0	301	0.30
Metastatic site		277	0.40
		251	0.50
Non-liver	0	223	0.60
Liver	74	191	0.70
Sunitinib dose (mg)		148	0.80
		79	0.90
25	72	14	0.95
37.5	81	Total point score	Probability of 3-year PFS
50	0	387	0.01
Hypertension		348	0.05
		324	0.10
No	78	292	0.20
Yes	0	265	0.30
Hand–foot syndrome		240	0.40
		215	0.50
No	100	187	0.60
Yes	0	154	0.70
		112	0.80
		43	0.90

## Data Availability

The data presented in this study are available on request from the corresponding author.

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
