# Peer review of "A Nomogram Predicting Progression Free Survival in Patients with Gastrointestinal Stromal Tumor Receiving Sunitinib: Incorporating Pre-Treatment and Post-Treatment Parameters"

_cancers, 2021, doi:10.3390/cancers13112587_

Round 1

Reviewer 1 Report

Main questions have been answered. However, some are left to be addressed:

1) I understand the explanations in the text regarding 299 and 109 patients. That is clear. Yet, I did not found Figure S1

2) Regarding the sunitinib concentrations. Please input the answer of the question (at least main idea) and the table in the main text. This data is broadening the originality of this paper.

3) Authors in the captions state fig. 1A is a progression-free survival (PFS) and fig. 1B overall survival (OS). These must be in the y axes in fig.1 A and fig. 1 B accordingly. Same goes in fig.3 B. The stated term „survival probability“ is confusing.

4) Fig 2 a and b fonts and resolution is still wrong.

Reviewer 2 Report

I do not have further comments

Author Response

Thanks

This manuscript is a resubmission of an earlier submission. The following is a list of the peer review reports and author responses from that submission.

Round 1

Reviewer 1 Report

The predictions on cancer growth are an important topic. Especially if the data is a clinical data. Therefore, this article is indeed relevant. However, some issues are in a need to be addressed.

Major issues.

  1. In the materials and methods (line 69) it was stated that in this paper 299 patients that fit the criteria for this study. However, in all other article the number was 109 patients. Can authors comment regarding these numbers?
  2. The patients were treated with two treatment designs. 1) 50 mg (4 weeks on and 2 weeks off); 2) continuously treated with 37,5 mg of sunitinib. I can try to understand that there was a reason in using two strategies. However, it is quite odd that these treatments were put in one treatment group. I have two points that support my statement.
    1. In first treatment strategy there were sequence 4 wk +2 we. So, in 6 weeks the total dose was (4wk*7days*50mg=1400mg). Strategy 2 there is (6wk*7days*1575mg). Therefore, Strategy 1 has 11,11 % higher overall dose.
    2. The daily dose during the treatment is EVEN 25 % higher. 50 mg vs. 37,5 mg.

3. Based on both observations, one can assume that results can differ. Could authors comment on that?

4. In the analysis 2.3 part (line 109) it is stated that samples were paraffin embedded and ect. What samples? Could authors broaden this part by stating what was the samples and how they were obtained?

5. ”For internal validation, bootstrapping with 1,000 resamples was used.” (Line 130) What are those “1,000 resamples”?

6. In figure 1 A and B authors show different survival probability on the same population when using the same Kaplan-Meier plot. In the captions authors claim that A part is a “progression-free survival (PFS)”. According to the National Cancer Institute NCI Dictionary of Cancer Terms the progression-free survival (PFS) is "the length of time during and after the treatment of a disease, such as cancer, that a patient lives with the disease but it does not get worse". To my humble understanding it means period of life when a patient lives with the disease but it does not get worse. When it gets worse then patient survives but its already not a PFS. Having this in mind fig.1 A part Y axis is termed wrong. It is not a survival probability.

7. Fig. 2 B is unreadable (too small). Please present in a clearer manner.

8. To my understanding (because stated in conclusions) pre-sunitinib sarcopenia and hypertension is a significant prognostic factor and a key data that is underlined as main find. If so it is plausible that it should be more broadly described not only in discussion but also in results. I can find it in a data tables. However, it should be underlined in some way.

9. Also if the hypertension is a key parameter it must be described how (with what strategy) it was measured to obtain significant results.

10. Authors state that their prediction model is quite accurate. As I understand the model was done on those 109 patients. If so, the model will be accurate on those 109 patients. However, it is not clear if it will be accurate for other patients than those 109 based on which the model was done. Have authors checked the model on other patients?

Minor issues

  1. “Sixty three male and 46 female” (Line 25). Please write only in numbers or only in letters.
  2. “trimmed to enrich tumor cellls.”

Author Response

Reviewer 1

The predictions on cancer growth are an important topic. Especially if the data is a clinical data. Therefore, this article is indeed relevant. However, some issues are in a need to be addressed.

Major issues.

  1. In the materials and methods (line 69) it was stated that in this paper 299 patients that fit the criteria for this study. However, in all other article the number was 109 patients. Can authors comment regarding these numbers?

Response 1: We found this statement is a little bit confusing. Therefore, we added a supplementary figure S1with the flow chart to illustrate the patient selection. We thought that this figure will help to realize the selected 109 patients from the original cohort 299 patients. Please see page 2, line 76 and Figure S1.

  1. The patients were treated with two treatment designs. 1) 50 mg (4 weeks on and 2 weeks off); 2) continuously treated with 37,5 mg of sunitinib. I can try to understand that there was a reason in using two strategies. However, it is quite odd that these treatments were put in one treatment group. I have two points that support my statement.
    1. In first treatment strategy there were sequence 4 wk +2 we. So, in 6 weeks the total dose was (4wk*7days*50mg=1400mg). Strategy 2 there is (6wk*7days*1575mg). Therefore, Strategy 1 has 11,11 % higher overall dose.
    2. The daily dose during the treatment is EVEN25 % higher. 50 mg vs. 37,5 mg.

Response 2: We thank to the reviewer’s comment. The two dosing strategies of sunitinib are considerably common used in real-world practice. Although there were no trials directly comparing the efficacy between these two dosing, comparisons across trials revealed that the results of daily 37.5mg in terms of both efficacy and toxicity overlapped with the phase III and early phase I/II patients (see Table below), suggesting that the two dosing might have similar efficacy and safety profiles. Consequently, sunitinib 37.5mg was thought as an alternative active regimen compared with the standard 50mg dose (4/2 schedule).

Phase I/II (Morgan, 2005)

Phase III (Demitri, 2006)

Phase II (George, 2009)

Eligible patients

After imatinib failure

Sunitinib dose

50mg (schedule 4/2)

37.5mg (fixed daily dose)

Design

Single arm (N=97)

Sunitinib (N=207) vs. placebo (N=105)

Morning (N=30) vs. Evening dose (N=30)

ORR

7%

7%

-

CBR

54%

58%

53%

PFS

7.8 months

27.3 weeks

34 weeks (around 8.5 months)

OS

19 months

73.9 weeks

107 weeks (around 26 months)

  1. Based on both observations, one can assume that results can differ. Could authors comment on that?

Response 3: As explained in the response 2, we thought that the two dosing were similarly effective in patients with GIST after imatinib failure. Our study is a single-arm retrospective analysis, therefore the two active regimens with comparable efficacy and safety profile were analyzed together as a whole.

  1. In the analysis 2.3 part (line 109) it is stated that samples were paraffin embedded and ect. What samples? Could authors broaden this part by stating what was the samples and how they were obtained?

Response 4: The specimen from biopsy or surgical resection of GIST with progression after imatinib use was processed with formalin-fixed and paraffin embedded. We added this sentence to broaden the sample information. Please see page 3, line 110-112.

  1. ”For internal validation, bootstrapping with 1,000 resamples was used.” (Line 130) What are those “1,000 resamples”?

Response 5: This statement is used to describe the bootstrapping approach. Briefly, resampling the raw data 1000 times by randomly selecting samples with replacement in this cohort.

  1. In figure 1 A and B authors show different survival probability on the same population when using the same Kaplan-Meier plot. In the captions authors claim that A part is a “progression-free survival (PFS)”. According to the National Cancer Institute NCI Dictionary of Cancer Terms the progression-free survival (PFS) is "the length of time during and after the treatment of a disease, such as cancer, that a patient lives with the disease but it does not get worse". To my humble understanding it means period of life when a patient lives with the disease but it does not get worse. When it gets worse then patient survives but its already not a PFS. Having this in mind fig.1 A part Y axis is termed wrong. It is not a survival probability.

Response 6: Our definition for PFS is exactly the same as the review’s comment. Regarding the survival plot such as figure 1A, the Kaplan-Meier estimate is the common way of computing the survival over time via a nonparametric approach. For each time interval, survival probability is calculated as the number of subjects surviving divided by the number of patients at risk. When graphing PFS or OS, probability of survival over time in percentage of patients treated (-- survival function, + censored) is illustrated in a step-down plot like below. This is an almost universally using tool for estimating survival analysis in cancer patients.

  1. 2 B is unreadable (too small). Please present in a clearer manner.

Response 7: We thank the reviewer for this suggestion. We re-produced this figure with high resolution.

  1. To my understanding (because stated in conclusions) pre-sunitinib sarcopenia and hypertension is a significant prognostic factor and a key data that is underlined as main find. If so it is plausible that it should be more broadly described not only in discussion but also in results. I can find it in a data tables. However, it should be underlined in some way.

Response 8: We agreed with your comment to highlight this part, especially in results. We put this part in results section, please see page 7, Section 3.2, line 182-184. We further created the nomogram using Cox proportional model analysis with 8 prognostic factors, including gender, ECOG, platelet/lymphocyte ratio, sarcopenia, metastatic site, sunitinib dose, hypertension, and HFSR.

  1. Also if the hypertension is a key parameter it must be described how (with what strategy) it was measured to obtain significant results.

Response 9: We measured the blood pressure at clinics visit every time. At every visit, we discussed with the patients about their blood pressure control and prescribing sunitinib with or without anti-hypertensive medications accordingly.

  1. Authors state that their prediction model is quite accurate. As I understand the model was done on those 109 patients. If so, the model will be accurate on those 109 patients. However, it is not clear if it will be accurate for other patients than those 109 based on which the model was done. Have authors checked the model on other patients?

Response 10: We appreciated this reviewer for this suggestion. In our study, we used the cohort as a whole to generate the prognostic model. We did not split into two sets for training set and test set for validation due to limited sample size. Therefore, this model was not tested in another independent cohort. We acknowledged and documented this limitation in the discussion section (Please see page 10, line 214-217). In line with this, in the conclusion section we also stated that our results need further external validation in the future. (Please see page 11, line 298-300)

Minor issues

  1. “Sixty three male and 46 female” (Line 25). Please write only in numbers or only in letters.

Response: Thank you. We changed them in letters.

  1. “trimmed to enrich tumor cellls.”

Response: We corrected the typo error. Please see page 3, line 110.

Reviewer 2 Report

Major

“Furthermore, we constructed a prognostic nomogram model using these significant pre-treatment and

post-treatment variables.” Nomograms have been developed to help clinicians to foresee the prognosis of a given patient and through this prognostication improving treatment choice. A nomogram using variables that will be known only after treatment is useless.

The aim of the study as expressed in the Introduction section, is “The present study aimed to establish a prognostic nomogram incorporating pre-treatment post-treatment factors and to predict PFS for advanced GIST patients receiving sunitinib”. Therefore, the developed nomogram predicts PFS obtained by sunitinib after having used sunitinib. This is purposeless.

Author Response

Reviewer 2

Major

“Furthermore, we constructed a prognostic nomogram model using these significant pre-treatment and post-treatment variables.” Nomograms have been developed to help clinicians to foresee the prognosis of a given patient and through this prognostication improving treatment choice. A nomogram using variables that will be known only after treatment is useless.

The aim of the study as expressed in the Introduction section, is “The present study aimed to establish a prognostic nomogram incorporating pre-treatment post-treatment factors and to predict PFS for advanced GIST patients receiving sunitinib”. Therefore, the developed nomogram predicts PFS obtained by sunitinib after having used sunitinib. This is purposeless.

Response: We thank to this reviewer for these comments. We understood the concern raised regarding post-treatment variables as part of prognostic model. The majority of variables in this model were baseline characteristics (pre-treatment). We included hypertension and hand-foot skin reaction (HFSR) as treatment-related variables in this model as well because they were significant from multivariate Cox regression model. How to optimally handle with the variables which may vary after treatment remains controversial from a statistics view. Some statistician advocated that variables after treatment were allowed to fit in the prognostic model as long as they were adjusted in a time-dependent manner, i.e., dynamic counted at each time interval of follow-up. Moreover, hypertension and HFSR have been found to significantly impact on prognosis from several studies focusing on inhibition of VEGFR pathway in colon cancer and renal cell cancer.  Another important reason using post-treatment AE (one month after sunitinib use) to predict response after use of sunitinib is the model is still useful and predictable before imaging evaluation using CT according to RECIST criteria. Therefore, from a clinical point of view, treatment-related adverse events still needs to take into consideration once they are related to prognosis.

Reviewer 3 Report

This is a well written manuscript about the outcome of GIST patients receiving second-line sunitinib. They developed a nomogram including both pre- and post-treatment parameters to predict response. In general, they describe interesting factors influencing the treatment effect as for example the role of sarcopenia status and BMI grading. This might be linked to the lower body weight people have in asia compared to US and EU patients. It could be useful to confirm their data and to check the relevance of their nomogram by validating it with a second patient cohort.

Throughout the manuscript many typos should be corrected.

Author Response

Reviewer 3

This is a well written manuscript about the outcome of GIST patients receiving second-line sunitinib. They developed a nomogram including both pre- and post-treatment parameters to predict response. In general, they describe interesting factors influencing the treatment effect as for example the role of sarcopenia status and BMI grading. This might be linked to the lower body weight people have in asia compared to US and EU patients. It could be useful to confirm their data and to check the relevance of their nomogram by validating it with a second patient cohort. Throughout the manuscript many typos should be corrected.

Response: We thank to this reviewer for these comments. We acknowledged this limitation and concluded this statement in the conclusion section. We look forward for the validation with another independent cohort. (Please see page 10, line 214-217 and page 11, line 298-300)

Reviewer 4 Report

Chang and colleagues present a well written, comprehensive, retrospective analysis of a cohort of 109 patients with metastatic GIST receiving palliative second line systemic therapy with sunitinib. Clinical and genetic characteristics are described, OS and PFS are presented, a nomogram to predict PFS is generated. Tables and a figures are included and nicely complement the written text.

I highly appreciate reading this article. Many thanks! Only a few comments:
  • PFS and OS are rather long compared to other available data. The fact that stagings after 3 years were performed only every 6 months (and not every 3 months) probably accounts to longer OS.
  • PDGFRA / SDH mutations would be of interest as they show higher clinical benefit of 2L sunitinib
  • was there a difference in OS and PFS observed depending on primary mutations?
  • why did you chose BMI cutoff of 27 (when adipositas is defined as >30)?

Author Response

Reviewer 4

Chang and colleagues present a well written, comprehensive, retrospective analysis of a cohort of 109 patients with metastatic GIST receiving palliative second line systemic therapy with sunitinib. Clinical and genetic characteristics are described, OS and PFS are presented, a nomogram to predict PFS is generated. Tables and a figures are included and nicely complement the written text.

I highly appreciate reading this article. Many thanks! Only a few comments:

  • PFS and OS are rather long compared to other available data. The fact that stagings after 3 years were performed only every 6 months (and not every 3 months) probably accounts to longer OS.

Response: Thanks to this reviewer for this comment. As the reviewer mentioned, the PFS and OS of our study were a bit longer than the reports from trials. The median PFS after sunitinib use is about 7-9 months in our study and other real-world analyses (please see the second paragraph of the discussion section). Therefore, we thought every 6 month follow-up after 3 years (36 months) might not substantially affect the estimation of PFS. The measurement of OS might be affected while the median OS is 31.8 months, indicating more than 50% patients had developed death events before 36 months of follow-up. Therefore, the overall survival function estimation for the majority of patients would not be largely biased.   

  • PDGFRA / SDH mutations would be of interest as they show higher clinical benefit of 2L sunitinib was there a difference in OS and PFS observed depending on primary mutations?

Response: To answer this interesting question, we performed this analysis to estimate the OS in different primary KIT mutation types. We did not found patients with wild-type GIST had superior OS. The overall comparison between exon 9 (N=11), exon 11(N=39), wild type (N=4), and others (N=2) was not significantly different in terms of OS (p=.197).

. why did you chose BMI cutoff of 27 (when adipositas is defined as >30)?

Response: We stratified the BMI according to the public education version of Ministry of Health and Welfare in Taiwan (https://www.hpa.gov.tw/Pages/Detail.aspx?nodeid=542&pid=9737). In this stratification system, BMI ≧27 is indicated as obesity. Our government recommended these criteria for adults to evaluate the presence of obesity (BMI ≧27). The reference is coming from one extension of Asian-Pacific consensus to define obesity. They suggested to lower BMI cut-off value from 30 to 27 for Asians according to literature evidence. (please see the reference below)

Wen-Harn Pan, Wen-Ting Yeh. How to define obesity? Evidence-based multiple action points for public awareness, screening, and treatment: an extension of Asian-Pacific recommendations. Asia Pac J Clin Nutr. 2008;17(3):370-4.

Round 2

Reviewer 2 Report

I am still concerned regarding including variables not statistically significant in the multivariate analyses.